# CMIMP: Effortlessly Achieving Diverse Population Training for Zero-Shot Coordination

## Abstract

Zero-shot coordination has recently become a hot topic in reinforcement learning research recently. It focuses on the generalization ability of agents, requiring them to coordinate well with collaborators that are not seen before without any fine-tuning. Population-based training has been proven to provide good zero-shot coordination performance; nevertheless, existing algorithms exhibit inefficiency, as the training cost scales linearly with the population size. To address this issue, this paper proposes the Conditional Mutual Information Maximized Population (CMIMP), an efficient training framework comprising two key components: a meta-agent that efficiently realizes a population by selectively sharing parameters across agents, and a mutual information regularizer that guarantees population diversity. To empirically validate the effectiveness of CMIMP, this paper evaluates it along with representational frameworks in Hanabi and confirms its superiority.

## 1 Introduction

Over these years, Multi-Agent Reinforcement Learning (MARL) has achieved remarkable success in various tasks, such as UAV navigation (Han et al., 2020), traffic signal control (Calvo & Dusparic, 2018) and resource allocation (Lin et al., 2018). To overcome the instability of reinforcement learning in multi-agent scenarios, researchers commonly adopt the strategy of self-play (Lowe et al., 2017), where a fixed group of agents are trained and tested together. This training paradigm endows agents with the capability to rapidly learn cooperative strategies, while posing the risk of overfitting to the training partners.

In order to improve generalization performance of cooperative agents, Hu et al. (2020) propose the problem of Zero-Shot Coordination (ZSC), which requires agents to coordinate with unknown agents without prior knowledge. One solution to this problem is reasoning about the task or partners (Shih et al., 2021; Li et al., 2023). Such solutions help agents learn consensus at the algorithm level, i.e. agents trained by the same framework can zero-shot coordinate well, but the agents still cannot coordinate well with agents trained by other types of algorithms (Lucas & Allen, 2022). Population-based training is another popular solution, which allows for training a best-response agent against a population of agents (Charakorn et al., 2022). One crucial advantage of these methods is that they can directly improve agents' zero-shot coordination performance with a population filled with diverse agents striving toward the same objective but exhibiting different behaviors. However, training a diverse population of agents significantly increases the computational cost. Simultaneously, there lack a robust and direct constraint to ensure that different agents act in distinct styles, while existing representative methods (Zhao et al., 2023) prefer to incorporate the average entropy of population actions carrying the risk of being misled by an agent exhibiting a random style. Besides, some population-training frameworks can only accommodate policies that output differentiable action distributions (Guo et al., 2024), hindering their practicality

Consequently, a new paradigm of population training for ZSC is needed that reduces computational costs while maintaining population diversity to achieve an efficient and diverse population. Meta-learning is an approach that enhances network's generalization ability through multi-task training. Inspired by that traditional meta-learning and multi-task learning (Tang et al., 2020; Kim & Sung, 2023) manage to train a single network with the ability of quickly adapting to various tasks while population training can also be treated as a multi task learning process, agent should also be able to perform various policies through a meta-policy and different task related adapters, thereby simulat-

Figure 1: The diagram of different training paradigms.

ing different agents with diverse policies in the population. Thus, a population can be constructed with a single meta-agent performing different policies to replace various population agents used in traditional methods, significantly reducing the computational costs for training diverse population agents while still providing various actions from diverse policies within the meta-agent.

Motivated by the aforementioned issues and studies, this paper introduces an efficient population-based training framework called Conditional Mutual Information Maximized Population (CMIMP). Within this framework, CMIMP leverages a meta-agent with hierarchical architecture, allowing multiple agents to share parameters for processing observations and history information, while using distinct sub-decision modules for action generation. This reduces the number of parameters and computational complexity by enabling N agents to generate actions in one single forward calculation. To ensure distinct behaviors among sub-decision modules, CMIMP maximizes the mutual information between actions and the sub-decision modules' index, conditioned on observations. Since some reinforcement learning (RL) frameworks cannot output differentiable action distributions, CMIMP optimizes an alternative objective: maximizing differences in preference values (e.g., Q-values or action selection probabilities) between sub-decision modules, which theoretically increases mutual information.

To empirically illustrate the effectiveness of CMIMP, we conduct experiments on Hanabi, a cooperative card game commonly used for zero-shot coordination research. For more comprehensive evaluation, we present two metrics: intra-algorithm cross-play scores (Hu et al., 2020) and one-sided zero-shot coordination cross-play scores (Lucas & Allen, 2022). CMIMP is evaluated along with several representative frameworks designed for zero-shot coordination and demonstrates superiority. Furthermore, we conduct additional experiments to explore the impact of specific settings on population training, including the pairing mode and population size.

Our contributions are summarized below:

- We analyze the necessity of population-based methods for zero-shot coordination and the limitations of existing methods in terms of computational complexity and population diversity. By comparing the similarities between population frameworks and meta-RL, we suggest that meta-RL can be employed to achieve an efficient population framework. In line with our motivation, we explore the application of meta-RL within this domain.
- We propose CMIMP, a novel zero-shot coordination framework consisting of a meta-agent with hierarchical architecture to realize a population and a conditional mutual information maximized scheme that guarantees population diversity by maximize the conditional mutual information of different action modules in meta-agent.
- We empirically validate the superiority of the proposed CMIMP over existing approaches in Hanabi. In comparison, our method achieves better zero-shot coordination performance, significantly enhances training efficiency, and reduces resource consumption. Moreover, we conduct ablation studies to investigate how different training modes affect the performance of population training.

## 2 RELATED WORK

Self-play (Yu et al., 2022) is a commonly-used technique in MARL, being able to quickly train a fixed group of cooperative agents while falling short in zero-shot coordination. To address this

Table 1: Performance comparison of different training paradigms

|  | Self-Play | Reasoning-based | Population-based | CMIMP (ours) |
|---|---|---|---|---|
| ZSC Performance | bad | medium | good | good |
| Training Speed | fast | slow $\sim$ fast | slow | fast |
| Versatility | bad $\sim$ good | bad $\sim$ good | medium | good |

issue, some researchers make agents reason about the task or partners. Related frameworks include breaking symmetries of the task to keep agents from learning specified strategies (Hu et al., 2020; Treutlein et al., 2021; Muglich et al., 2022), conducting multi-level reasoning to get higher-level consensus (Cui et al., 2021; Hu et al., 2021) and requiring agents to predict partners' actions (Lucas & Allen, 2022; Yan et al., 2024). The core disadvantage of the above solutions is that agents might form an algorithm-level consensus or overfit to several partners.

Another kind of mainstream solution is population-based training. This solution improves the generalization ability of a main agent by requiring it to cooperate well with all partner agents in a population. Consequently, improving the divergence of the population becomes a primary objective. Typical frameworks include reducing the collaboration scores of different partner agents within the population to make them behave differently (Charakorn et al., 2022; Rahman et al., 2023), improving trajectory diversity of different agents (Lupu et al., 2021) and increasing policy entropy of the population (Zhao et al., 2023). However, this kind of framework may face heavy computational load or limited usage. For example, most population training frameworks train distinct neural network parameters for agents in a population, which is time-consuming; Several frameworks (Lupu et al., 2021; Zhao et al., 2023) require agents to output differentiable action distribution, while some important RL frameworks such as value-based methods may not meet the requirement. In comparison, our proposed CMIMP utilizes a meta-agent to efficiently achieve population training, and designs a generic mutual information term to guarantee population divergence. A brief feature comparison of the aforementioned solutions are presented in Table 1.

Notably, the meta-agent in CMIMP is different from the agents in meta-RL (Nagabandi et al., 2018; Gupta et al., 2018). Traditional meta-RL aims to train agents that can quickly adapt to various tasks, the diversity of which is innate and invariable. In contrast, CMIMP's meta-agent is designed to exhibit various policies, the diversity of which is variable and what we seek to augment.

## 3 CONDITIONAL MUTUAL INFORMATION MAXIMIZED POPULATION FRAMEWORK

In order to achieve efficient and versatile population training for zero-shot coordination, we propose a novel CMIMP framework consisting of a hierarchical meta-agent that efficiently realizes population and a conditional mutual information maximization term that guarantees population diversity. Notably, the term is versatile since it does not require agents to output differentiable action distributions like some related works (Lupu et al., 2021) do. The details of CMIMP are presented below.

### 3.1 HIERARCHICAL META-AGENT FOR EFFICIENT POPULATION

One major drawback of population training is that it has to train multiple agents, which is time-consuming. Considering that meta-RL techniques manage to train meta-agents with the ability of quickly adapting to various tasks, selectively sharing parameters across different agents in a population is feasible: those task-related parameters (correspond to modules that process observation and keep history memory) can be shared, and those behavior-related parameters (correspond to modules used for decision-making) can be individually optimized.

Based on the aforementioned idea, we design a meta-agent with hierarchical architectures to realize a population. It consists of several neural-network-based modules, including an observation encoder $f^o$ that processes observations, an LSTM $f^l$ that keeps historical information and updates hidden states, a value head $f^v$ that outputs state value $v^t$, and several sub-decision modules $f^{a1}, f^{a2}, ..., f^{aK}$ that output vectors $q^t_{a1}, q^t_{a2}, ..., q^t_{aK}$ used for choosing actions. Here $q^t_{ai}$ is treated

as the policy output for the $i$-th agent in the population, and $K$ is the population size. Besides, the meta-agent is compatible with value-based and policy-gradient-based training paradigms, and the corresponding outputs of sub-decision modules are Q-values or action distributions. The above calculation is formulated below:

$$
\begin{aligned}
h^t, c^t &= f^l(h^{t-1}, c^{t-1}, f^o(o^t)) \\
v^t &= f^v(h^t) \\
q_{ai}^t &= f^{ai}(h^t) \, \forall i \in \{1, 2, ..., K\}
\end{aligned}
\tag{1}
$$

As can be seen, the hierarchical architecture of the meta-agent greatly reduces the parameters that need to be optimized (N complete sets $\rightarrow$ 1 complete sets + N subsets), thereby reducing the number of required interactions with the environment and accelerating training.

## 3.2 CONDITIONAL MUTUAL INFORMATION MAXIMIZATION

If all the agents in a population have similar policies, the best-response agent can only learn to co-operate with partners of one kind of policy, and in this way the advantage of population training disappears (See Sec. 4.4 for details). Therefore, increasing the diversity of population (i.e. different agents in the population behave variously) has always been a key concern in this research area. It is noteworthy that a diverse population means different agents in the population act differently, and this can be achieved by making the meta-agent output distinct actions with different sub-decision modules given an observation and history trajectory. This operation can be formulated as maximizing the conditional mutual information:

$$
I(A; U|H) = \int \int \int p(a, u, h) \log \frac{p(h)p(a, u, h)}{p(u, h)p(a, h)} du\,ds\,da
\tag{2}
$$

where $H$ represents observation input (containing current observation and historical trajectory used for decision), $U$ represents the index of sub-decision module that outputs action $A$, and $p(a, u, h)$ is the joint probability density function. Considering that reinforcement learning frameworks commonly estimate integrals using the Monte Carlo method, that is, sampling transitions from replay buffer to calculate, the unbiased estimation of mutual information $\hat{I}(A; U|H)$ can be written as:

$$
\begin{aligned}
\hat{I}(A; U|H) &= \frac{1}{N} \sum_{j=1}^{N} \log \frac{p(a_j|u_j, h_j)}{p(a_j|h_j)} \\
&= \frac{1}{N} \sum_{j=1}^{N} \left[ \log p(a_j|u_j, h_j) - \log \sum_{i=1}^{K} p(u_i|h_j)p(a_j|u_i, h_j) \right]
\end{aligned}
\tag{3}
$$

where $K$ is the total number of sub-decision modules, $N$ is the number of transitions, $a_j, u_j, h_j$ are action, sub-decision module index and observation input of the $j$-th transition, and $p(a_j|u_j, h_j)$ is the conditional probability that an agent takes action $a_j$ given $u_j, h_j$. For brevity, use $I_j$ to denote the $j$-th term in $\hat{I}(A; U|H)$:

$$
I_j := \log p(a_j|u_j, h_j) - \log \sum_{i=1}^{K} p(u_i|h_j)p(a_j|u_i, h_j)
\tag{4}
$$

Notably, directly maximizing $\hat{I}(A; U|H)$ with gradient-based methods is not a preferable choice for two reasons. Firstly, The posterior probability $p(u_i|h_j)$ is hard to calculate. Secondly, the gradient of $p(a_j|u_i, h_j)$ is almost always equal to zero for many RL policies. For example, value-based policies commonly output actions that maximize the action-value function $Q(a, u, h)$ (or use $\epsilon$-greedy for exploration). In this way, the value of $p(a_j|u_i, h_j)$ is determined only by whether $a_j = \arg\max_a Q(a, u_i, h_j)$. Consequently, the derivatives of $p(a_j|u_i, h_j)$ with respect to the neural network parameters are equal to zero.

We propose to optimize an alternative objective $\bar{I}(A; U|H)$, which have the following two properties:

1. $\bar{I}(A; U|H)$ provides gradients that can be used for neural network training, whether the meta-agent makes decisions by outputting action distributions or maximizing Q-functions.

2. Increasing $\bar{I}(A;U|H)$ also increases $\hat{I}(A;U|H)$.

The definition of $\bar{I}(A;U|H)$ is given below:

$$\bar{I}(A;U|H) = -\frac{1}{N} \sum_{j=1}^{N} \sum_{i=1, i \neq j}^{K} F(u_i, h_j, a_j) \tag{5}$$

where $F(u_i, h_j, a_j)$ represents the favor of the meta-agent for $a_j$ given $u_i, h_j$ and is required to be the direct output of a neural network so that its gradients can be used for gradient-based training. Below are several possible forms of $F(u_i, h_j, a_j)$:

1. If the neural network used for decision directly outputs action distribution (common for policy-gradient-based methods such as PPO (Schulman et al., 2017)), then $F(u_i, h_j, a_j)$ represents the probability of choosing $a_j$ given $u_i, h_j$, which is $p(a_j|u_i, h_j)$;

2. If the neural network used for decision outputs advantage functions (or Q-values, common for value-based methods such as Dueling-DQN (Wang et al., 2016)), then $F(u_i, h_j, a_j) = A(u_i, h_j, a_j)$ (or $Q(u_i, h_j, a_j)$).

Moreover, we provide certain theoretical guarantee for maximizing $\bar{I}(A;U|H)$.

**Theorem 1.** *Given $F(u_j, h_j, a_j)$, if $F$ is update to $F'$ such that:*

$$\exists v \ s.t. \ \arg\max_a F(u_v, h_j, a) = a_j \ \wedge \ F'(u_v, h_j, a_j) < F(u_v, h_j, a_j) \tag{6}$$
$$\forall i \neq v, \ F'(u_i, h_j, a_j) = F(u_i, h_j, a_j)$$

*then the corresponding term $I_j$ in $\hat{I}(A;U|H)$ is updated to $I'_j$ and satisfies $I'_j \geq I_j$.*

The proof is presented in the Appendix.

Calculating $\bar{I}(A;U|H)$ requires obtaining the action outputs of all the agents in the population under the same observation input, which can be efficiently done with the meta-agent: only one forward calculation provides the required outputs. In comparison, this operation will be quite time-consuming in typical population training as $N$ forward calculations are needed, especially given that LSTM forward propagation is slow.

We would like to emphasize that the two components of CMIMP operate in a complementary manner. The meta-agent constitutes the fundamental network architecture, laying a good foundation for efficient computation of mutual information and training. Concurrently, the conditional mutual information term serves as an imperative guiding force throughout the training process, ensuring the production of a diverse meta-agent.

### 3.3 INSTANTIATION

CMIMP only specifies how to build an efficient and diverse population, and is compatible with multiple base RL frameworks that optimize agent policies. Our instantiation is based on an value-based approach because it is confirmed that this kind of method is suitable for our experimental task Hanabi (Hu & Foerster, 2019; Bard et al., 2020). The main agent only needs to cooperate well with all the agents in the population (which are realized using the partner meta-agent), and thus it is required to minimize the base TD-error (Van Hasselt et al., 2016):

$$L_m = \frac{1}{N} \sum_{j=1}^{N} r_j + \gamma \max_a Q_{\theta'_m}(h'_j, u_j, a) - Q_{\theta_m}(h_j, u_j, a_j) \tag{7}$$

where $\theta_m$ and $\theta'_m$ represent the parameters in the online Q-net and target Q-net of the main agent respectively. Notably, the main agent has the same neural network architecture as a normal agent shown in Fig. 1. In comparison, the meta-agent not only needs to learn coordination, but also needs to become diverse by maximizing $\bar{I}(A;U|H)$. Consequently, $\bar{I}(A;U|H)$ is added to the base TD-loss with a weight $\alpha$, which controls the balance between cooperation ability and population diversity:

$$L_p = \frac{1}{N} \sum_{j=1}^{N} \left[ r_j + \gamma \max_a Q_{\theta'_p}(u_j, h'_j, a) - Q_{\theta_p}(u_j, h_j, a_j) + \alpha \sum_{i=1, i \neq j}^{K} Q_{\theta_p}(u_i, h_j, a_j) \right] \tag{8}$$

Table 2: Feasible population training modes

| Index | Act Group | Optimization Objective for $\pi_m$ | Optimization Objective for $\pi_{pi}$ |
|---|---|---|---|
| I | $MP$ | $\sum_{i=1}^{N} J(\pi_m, \pi_{pi})$ | $J(\pi_m, \pi_{pi})$ |
| II | $MM, MP$ | $J(\pi_m, \pi_m) + \sum_{i=1}^{N} J(\pi_m, \pi_{pi})$ | $J(\pi_m, \pi_{pi})$ |
| III | $MP, PP$ | $\sum_{i=1}^{N} J(\pi_m, \pi_{pi})$ | $J(\pi_{pi}, \pi_{pi})$ |
| IV | $MM, MP, PP$ | $J(\pi_m, \pi_m) + \sum_{i=1}^{N} J(\pi_m, \pi_{pi})$ | $J(\pi_{pi}, \pi_{pi})$ |
| V | $MP, PP$ | $\sum_{i=1}^{N} J(\pi_m, \pi_{pi})$ | $J(\pi_{pi}, \pi_{pi}) + J(\pi_m, \pi_{pi})$ |
| VI | $MM, MP, PP$ | $J(\pi_m, \pi_m) + \sum_{i=1}^{N} J(\pi_m, \pi_{pi})$ | $J(\pi_{pi}, \pi_{pi}) + J(\pi_m, \pi_{pi})$ |

where $\theta_p$ represent the parameters in the online Q-net of the partner meta-agent. To accelerate convergence, prioritized replay (Schaul et al., 2015) and dueling-net (Wang et al., 2016) is also utilized.

Another key component of instantiation is the training mode, which specifies the kind of pairs of agents are used to interact with the environment and corresponding transitions used for training agents. Table 2 summarizes six feasible training modes and details are presented in the Appendix. Take Mode-III as an example: it has act groups $MP, PP$, which means agents interact with the environment and generate transitions in two groups: [main agent, partner agent] and [partner agent, partner agent]. Besides, the training objectives require the main agent to cooperate well with the partner agent, while the partner agent only needs to optimize self-play scores and needs not to adapt to the main agent. Each training mode has its own emphasis, and we investigate the performance of different training modes in Sec. 4.5.

## 4 EXPERIMENTS

### 4.1 EXPERIMENTAL ENVIRONMENT

We conduct experiments on Hanabi (Bard et al., 2020), a card game which requires players to co-operatively play cards of different colors and ranks in order. Playing cards wrongly leads to the loss of life tokens, and the shared team score is the number of cards that have been correctly played at the end of the game. Notably, players can only view the cards of the collaborators, yet lack the capacity to observe the cards in their own hands. Due to this setting, players have to reason others' intention as well as convey information through actions. Therefore, self-play agents can easily get high scores for they are familiar with each other, while cooperating with strangers gets quite hard, making Hanabi a popular benchmark for zero-shot coordination research.

### 4.2 EVALUATION CRITERIA FOR ZERO-SHOT COORDINATION

Zero-shot coordination requires agents to cooperate well with collaborators that are not seen before, namely "strangers". Since "strangers" does not refer to specific datasets or agents, the corresponding evaluation criteria are slightly different from those of normal MARL. Below we give a formulaic representation in two-agent scenarios.

Use $J(\pi_1, \pi_2)$ to denote the expected cumulative discounted return obtained by the collaboration of $\pi_1$ and $\pi_2$. Use $\pi_i^M$ to denote the policy obtained with training framework $M$ and random seed $i$. The earliest metric to evaluate zero-shot cooperation performance is intra-algorithm cross-play (abbreviated as Intra-XP) score (Hu et al., 2020):

$$S_{intra-XP}(M) = \mathbb{E}[J(\pi_i^M, \pi_j^M)|i \neq j] \tag{9}$$

Cross-play score represents how well agents cooperate with partners that are trained by the same framework but different seeds. This metric is easily accessible and relatively objective, but has a strong assumption on the "strangers". Besides, considering that testing partners are trained with the same algorithm, the main test agent has a little prior information about them, therefore, it is not rigorous to judge zero-shot coordination performance based on this metric alone.

Table 3: Zero-shot coordination performance of different frameworks

|  | SP | OP | OBL | TrajeDi | MEP | CMIMP |
|---|---|---|---|---|---|---|
| Intra-XP | 3.96±0.49 | 14.94±0.67 | **23.80±0.03** | 12.95±1.25 | 20.48±0.16 | 21.05±0.05 |
| 1ZSC-XP | 7.68±0.39 | 13.48±0.19 | 3.80±0.07 | 12.92±0.36 | 14.76±0.11 | **15.73±0.03** |

Table 4: Training costs of different population-based training frameworks with population size 5

|  | TrajeDi | MEP | CMIMP |
|---|---|---|---|
| Training time of 500 epochs(days) | 5.40 | 5.23 | **0.92** |
| Memory usage(GB) | 158.38±2.07 | 161.18±1.31 | **54.93±2.76** |

In order to address the deficiencies of the aforementioned metric, Lucas & Allen (2022) propose one-sided zero-shot coordination (abbreviated as 1ZSC-XP) score:

$$S_{1ZSC-XP}(M) = \mathbb{E}[J(\pi^M, \pi^{M_t})] \tag{10}$$

where $M_t$ refers to a set of algorithms that are not specially designed for zero-shot coordination. The shortcoming of this criterion is that $\pi^{M_t}$ still cannot represent all feasible "strangers", and the results may be biased.

Neither of the metrics is perfect, hence the following sections display the above two metrics for more comprehensive evaluation.

### 4.3 COMPARATIVE EXPERIMENTS

To empirically validate the superiority of CMIMP, we test it along with the following frameworks:

**SP**(Self-Play): The baseline self-play training with parameter sharing which acts as a baseline.

**OP** (Hu et al., 2020): A classical framework that improves zero-shot coordination by breaking symmetries in self-play training.

**OBL** (Hu et al., 2021): A representational framework that trains policies with multi-level cognitive reasoning and thus avoids over-fitting to certain training partners.

**TrajeDi** (Lupu et al., 2021): A population-based training framework that improves the trajectory diversity of agents in the population.

**MEP** (Zhao et al., 2023): A population-based framework that improves diversity by maximizing the average policy entropy of the population.

For all population-based methods (TrajeDi, MEP and CMIMP), the population is realized using our proposed meta-agent with population size 5 for ease of comparison while we also replicate the common population frameworks of TrajeDi and MEP for comparative analysis of training efficiency. Besides, TrajeDi and MEP require agents to output differentiable action distribution and thus act in a Boltzmann way[1] instead of the conventional $\epsilon$-greedy way.

To enhance the credibility of the evaluation, we train five models with different random seeds under each framework, and test Intra-XP and 1ZSC-XP scores introduced in Sec.4.2. Specifically, 1ZSC-XP scores are obtained by pairing the tested zero-shot coordination agents with 40 non-ZSC agents obtained with four kinds of self-play frameworks: IQL (Tan, 1993), VDN (Sunehag et al., 2018), SAD and SAD+AUX (Hu & Foerster, 2019).

Tab. 3 shows the mean and standard error of the evaluation while Tab. 4 compare the training cost of different representative population-based training methods. It can be concluded that CMIMP has the best zero-shot coordination performance and shows significantly efficiency and lightweight performance improvement: it scores the highest in 1ZSC-XP, and its Intra-XP score is only second to OBL. But this does not imply that OBL has better zero-shot coordination performance for the 1ZSC-XP score of OBL is significantly lower than the Intra-XP score, which may because OBL

---

[1]Sampling actions from a distribution obtained with $SoftMax(Q)$.

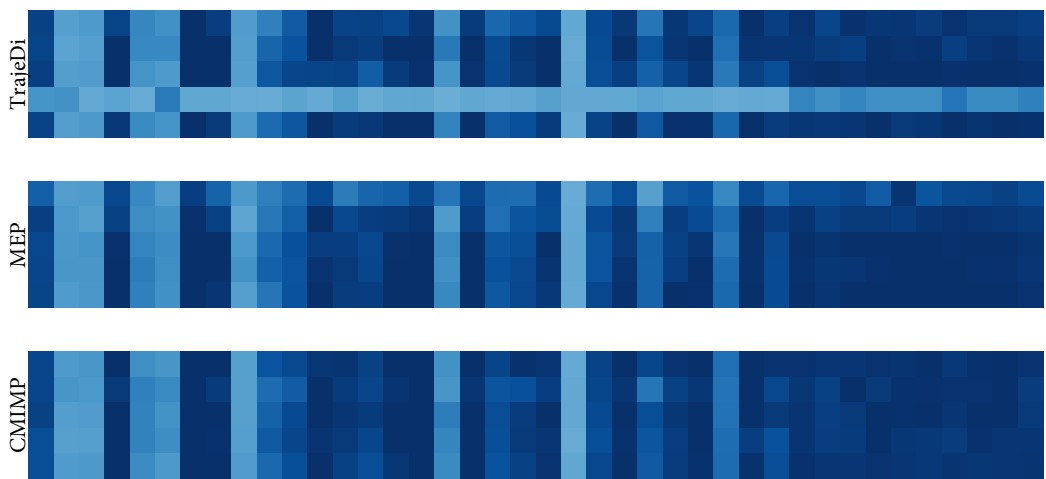

Figure 2: Detailed pair-wise 1ZSC-XP scores of TrajeDi, MEP and CMIMP. Deeper colors represent higher scores and each row represents the coordination scores of testing a main agent pairing with 40 non-ZSC agent, thus forming a $5 \times 40$ heat-map.

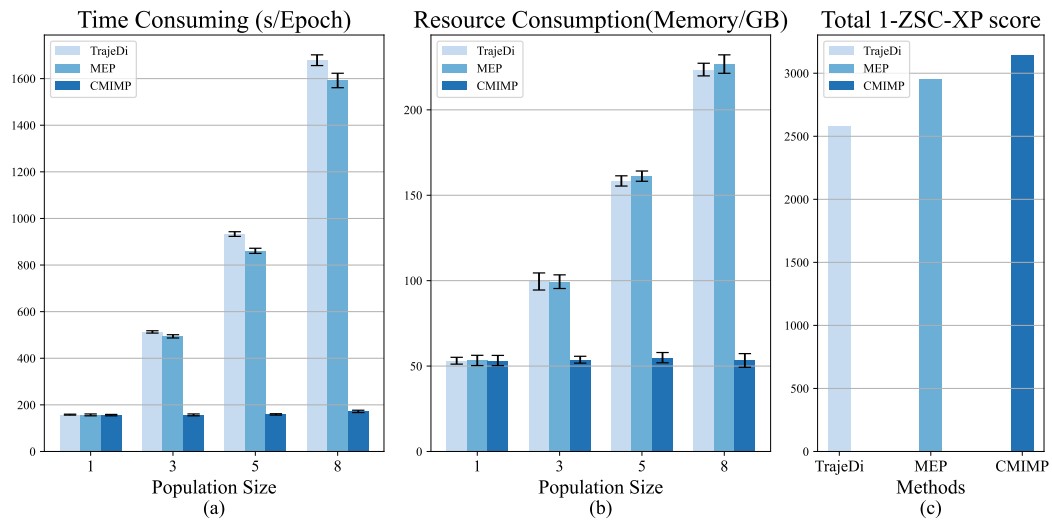

Figure 3: Comparison of different population-based frameworks : (a) training time consuming with population size increase; (b) resource consumption with population size increase; (c) total score of pair-wise 1ZSC-XP scores from Fig. 2.

forms an algorithm-level consensus to some extent, making OBL agents incompatible with agents trained with other algorithms.The polarized performance of OBL on these two metrics also supports our previous analysis on the inadequacy of only using the Intra-XP metric. OP, TrajeDi and MEP can more or less improve zero-shot coordination compared to the baseline SP, but are all inferior to CMIMP. Compared to training efficiency and resource requirement in the condition of population size 5, CMIMP achieves 5.77 times the training efficiency and only require one-third of the memory resources, which can demonstrate the significant improvement in addressing the huge resource consumption of existing methods while improving zero-shot coordination performance among agents.

Besides, Fig. 2 visualizes the detailed pair-wise 1ZSC-XP cooperation scores of CMIMP using a heat-map. For ease of comparison, Fig. 3(c) shows the total score of heat-maps. The heat-maps

of three compared population-based methods are presented along for reference, and the heat-maps of other frameworks are presented in the Appendix. In each sub-figure, each row represents the coordination scores of a testing main agent pairing with all non-ZSC agents, and deeper colors represent higher scores. Different rows in a sub-figure correspond to agents trained with different random number seeds under the same framework. There are two phenomena worth noting: Firstly, the differences between columns are consistent for these two frameworks. The reason is that some non-ZSC agents are relatively easy to cooperate with (e.g. column 7,8), while some others are not (e.g. column 2,3). Secondly, the performance stability of CMIMP is good: 1ZSC-XP scores of CMIMP models with different random seeds vary little, while TrajeDi and MEP demonstrate certain instability.

To further compare the performance and efficiency of these three methods, among these three methods, Fig. 3 illustrates the differences in training duration and resource consumption as the population size increases between TrajeDi, MEP using the original framework, and CIMIP utilizing a meta-agent population. The results indicate that CIMIP is largely unaffected by the population size, demonstrating its flexibility to adapt to complex tasks requiring larger population scales( i.e. tasks with huge action space). In contrast, traditional population frameworks exhibit a nearly linear increase in training costs with the growth of population size, rendering them less feasible for large-scale population training.

## 4.4 ABLATION STUDY

As is stated in (8), the training objective for the meta-agent has a weighted mutual-information term that makes the sub-decision modules of the meta-agent act differently. Then how will this term affect the training process? Fig. 4 shows the variation curves of the following four metrics over training epochs under different training modes:

**MM Score**: The self-play score of the main agent;

**MP Score**: The cooperation score of the main agent and the partner agent, one of the key objectives of population training;

**PP Score**: The self-play score of the partner agents;

**Diff Prob**: The probability of different partner agents in the population choosing the same actions under the same observation input. This metric for a diverse population should be relatively low.

As is shown in Fig. 4, when the mutual information term $\bar{I}(A;U|H)$ is ignored (i.e. $\alpha = 0$), **Diff Prob** quickly rises to 1, meaning that the partner agents in the population act similarly. As a result, the generalization performance of the main agent is reduced (low XP scores during testing) despite that the training process goes smoothly (high MM/MP/PP scores during training). When $\alpha$ is set to a proper value ($\alpha = 1$), **Diff Prob** maintains relatively low, indicating a diverse population. When $\alpha$ is set to a large value ($\alpha = 10$), the population diversity does not further increase, and what's worse, the self-play score of partner agents (PP score) goes low. This indicates that the rationality of the partner behavior may be affected due to large $\alpha$, and the coordination performance of the main agent is also hampered. To sum up, $\bar{I}(A;U|H)$ can help build a diverse population and thereby improve the zero-shot coordination, while assigning it a too large weight might have negative effect.

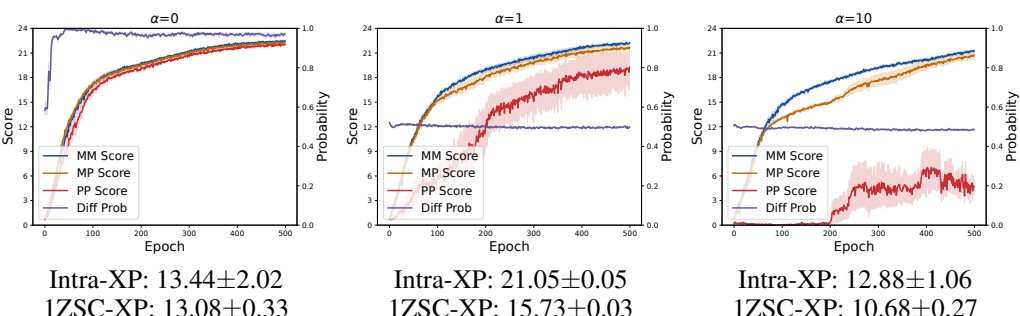

Intra-XP: 13.44±2.02          Intra-XP: 21.05±0.05          Intra-XP: 12.88±1.06
1ZSC-XP: 13.08±0.33          1ZSC-XP: 15.73±0.03          1ZSC-XP: 10.68±0.27

Figure 4: Training curves and testing scores of CMIMP with different $\alpha$.

Table 5: Zero-shot coordination performance evaluation of different training modes

|          | I         | II              | III        | IV         | V          | VI         |
| -------- | --------- | --------------- | ---------- | ---------- | ---------- | ---------- |
| Intra-XP | 1.10±0.13 | **21.05±0.05**  | 8.58±1.76  | 12.56±1.37 | 12.31±1.65 | 19.18±0.30 |
| 1ZSC-XP  | 5.72±0.16 | **15.73±0.03**  | 10.41±0.36 | 12.19±0.24 | 12.41±0.28 | 15.10±0.04 |

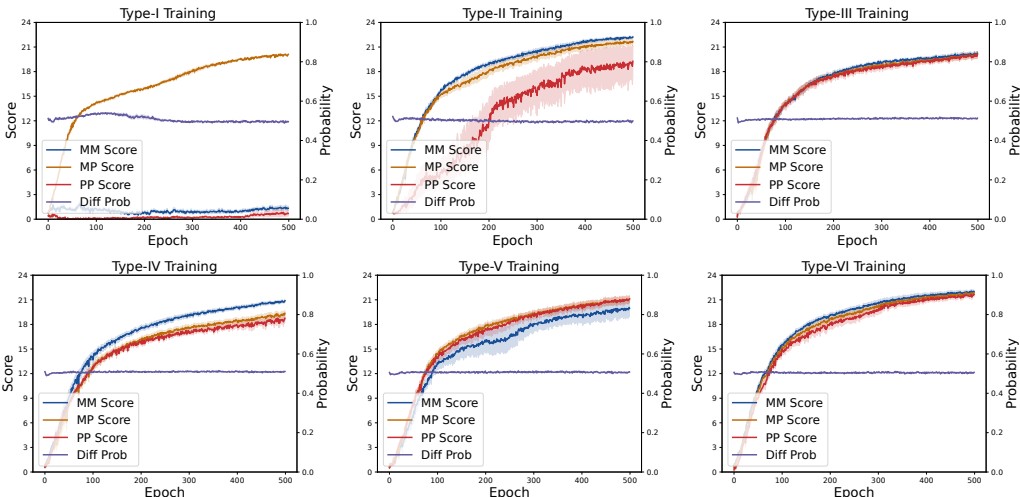

Figure 5: Training curves of different training modes.

### 4.5 COMPARISON OF DIFFERENT TRAINING MODES

Table 2 introduces six feasible training modes for population-based training. Then which mode is the best? We present training curves of different modes in Fig. 5. It can be seen that **Diff Prob** and **MP Score** exhibit consistent trends across all training modes, indicating that different modes are consistent in optimizing the primary objective ($J(\pi_m, \pi_p)$) and enhancing diversity. However, Tab. 5 confirms that zero-shot coordination performance of different training modes varies a lot, and such differences are brought by the settings of secondary objectives. Mode-I is the worst, indicating that only optimizing the primary objective ($J(\pi_m, \pi_p)$) is not enough. Mode-II is the best, confirming the necessity of adding the self-play objective for the main agent. Notably, Mode-IV and Mode-VI additionally require increasing self-play scores for partner agents on the basis of Mode-II, and this operation is of no benefit judging from the results.

## 5 CONCLUSION

In this paper, we discuss the necessity of population training for the zero-shot coordination and highlight the logical commonalities between population training and meta-RL or multi-task learning, which can address the inefficiencies of existing population-based zero-shot coordination methods due to outdated training frameworks. Driven by this motivation, we propose an efficient population-based zero-shot coordination framework, called CMIMP, to achieve a simulation of diverse populations of any population size through a single parameter adjustment, while incurring almost no additional training costs compared to training a single agent. Experiments conducted in Hanabi validate the outstanding performance of our proposed method in zero-shot coordination capabilities, efficiency, and low resource requirements. Additionally, our proposed training framework demonstrates promising potential for large-scale population training, as its training is unaffected by population size, allowing for the implementation of extremely large-scale population training, which could further enhance zero-shot coordination capabilities.

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

## A  PROOF OF THEOREM 1

*Proof.* In consideration of the relationship between $F(u, h, a)$ and $p(u, h, a)$, there are two cases to be addressed.

**Case 1**  $F(u, h, a) = p(a|u, h)$

With the condition stated in (6), only $p(a_j|u_v, h_j)$ will be changed among all the terms in $I_j$. Consequently,

$$
\begin{aligned}
I'_j - I_j = \log &\left[ p(u_v|h_j)p(a_j|u_v, h_j) + \sum_{i=1, i \neq v}^{K} p(u_i|h_j)p(a_j|u_i, h_j) \right] \\
- \log &\left[ p(u_v|h_j)p'(a_j|u_v, h_j) + \sum_{i=1, i \neq v}^{K} p(u_i|h_j)p(a_j|u_i, h_j) \right]
\end{aligned}
\tag{11}
$$

Since $p'(a_j|u_v, h_j) < p(a_j|u_v, h_j)$ and $p(u_v|h_j) \geq 0$, $I'_j \geq I_j$.

**Case 2**  $F(u, h, a) = A(u, h, a)$

In this case, $p(a|u, h) = 1$ if $a = \arg\max Q(u, h, a)$ where $Q(u, h, a) = A(u, h, a) + V(u, h)$, else $p(a|u, h) = 0$ [2].

According to (6), only $Q(u_v, h_j, a_j)$ changes, and this leads to three possible outcomes:

1. $a_j \neq \arg\max Q(u_v, h_j, a)$ and $a_j \neq \arg\max Q'(u_v, h_j, a)$. In this situation, $p(a_j|u_v, h_j)$ remains the same, and $I'_j = I_j$.

2. $a_j = \arg\max Q(u_v, h_j, a)$ and $a_j = \arg\max Q'(u_v, h_j, a)$. Similarly, $I'_j = I_j$.

3. $a_j = \arg\max Q(u_v, h_j, a)$ and $a_j \neq \arg\max Q'(u_v, h_j, a)$. In this situation, $p(a_j|u_v, h_j) = 1$ and $p'(a_j|u_v, h_j) = 0$. As is proved before, $p'(a_j|u_v, h_j) < p(a_j|u_v, h_j)$ leads to $I'_j \geq I_j$.

$\square$

## B  IMPLEMENTATION DETAILS

**Hardware and software settings**  We experiment on a server with 2 x Tesla P100 and a Intel Xeon Platium CPU (12 cores), and training one models takes around 15 hours. The experimental codes are modified based on the open source codes of OBL (Hu et al., 2021).

**Neural network hyper parameters**  (1) formulates the decision process, and below introduces the hyper parameters of each module. $f^o$ is a linear transform with output size 512. LSTM $f^l$ has two layers with hidden dim 512. $f^v$ is a linear transform with output size 1, and $f^{ai}$ are linear transform matrix with output size equaling action dim.

**Training hyper parameters**  All the models are training 500 epochs with replay buffer size 50000 and batch size 128. Parameters are updated via Adam optimizer with learning rate 6.25e-5. Discount factor $\gamma$ is set to 0.999.

---

[2]If the meta-agent uses an $\epsilon$-greedy strategy for exploration, then the corresponding value is $1 - \epsilon$ and $\epsilon/(|A| - 1)$. This difference has no impact on the proof.

## C  DETAILED INTRODUCTION OF DIFFERENT TRAINING MODES

Table. 2 presents several population training modes, and the following content takes Mode IV as an example to introduce the meaning of each column:

- **Act Group: MM, MP, PP.**: There are three kinds of act groups will be used for interacting with the environment and generating transitions: [Main agent, Main agent], [Main agent, Partner agent] and [Partner agent, Partner agent]. Consequently,

- **Optimization Objective for** $\pi_m$**:** $J(\pi_m, \pi_m) + \sum_{i=1}^{N} J(\pi_m, \pi_{pi})$: The main agent is required to cooperate well with itself and partner agents. This influences the transitions used for training main agent: it has two kinds of transitions, which are playing records with another main agent and playing records with a partner agent, and both of them are used for calculating main agent loss defined in (7).

- **Optimization Objective for** $\pi_p$**:** $J(\pi_{pi}, \pi_{pi})$: The partner agents are only required to cooperate well with itself. Notably, it has two kinds of transitions, which are playing records with another partner agent and playing records with a main agent, and only the first will be used for calculating partner agent loss defined in (8). In contrast, in Mode VI, two kinds of transitions are both used for training due to the different optimization objective for $\pi_p$.

Algorithm 1 introduces the training process.

---

**Algorithm 1** Training process of CMIMP with Mode IV

**INPUT:** Mutual information term weight $\alpha$, batch size $N_b$, replay buffers $A$, $B$;
1: Initialize $\theta \leftarrow$ random, $\theta_p \leftarrow$ random;
2: Define action groups: $G_1 = [\text{Main agent}, \text{Main agent}], G_2 = [\text{Main agent}, \text{Partner agent}], G_3 = [\text{Partner agent}, \text{Partner agent}]$;
3: **while** not reached maximum iterations **do**
4:    **for** $G \in \{G_1, G_2, G_3\}$ **do**
5:       Reset environment if necessary;
6:       $o_1^t, o_2^t \leftarrow \text{Observe}(G)$;
7:       $h_1^t, h_2^t \leftarrow \text{Update\_hidden\_states}(o_1^t, o_2^t)$;
8:       $a_1^t \leftarrow \pi_\theta(h_1^t), a_2^t \leftarrow \pi_\theta(h_2^t)$;
9:       $r_1^t, r_2^t \leftarrow \text{Environment\_rewards}(a_1^t, a_2^t)$;
10:      **if** $G = G_1$ **then**
11:        Store $(o_1^t, h_1^t, a_1^t, r_1^t, o_1^{t+1})$ and $(o_2^t, h_2^t, a_2^t, r_2^t, o_2^{t+1}) \in A$;
12:      **end if**
13:      **if** $G = G_2$ **then**
14:        Store $(o_1^t, h_1^t, a_1^t, r_1^t, o_1^{t+1}) \in A$;
15:      **end if**
16:      **if** $G = G_3$ **then**
17:        Store $(o_1^t, h_1^t, a_1^t, r_1^t, o_1^{t+1})$ and $(o_2^t, h_2^t, a_2^t, r_2^t, o_2^{t+1}) \in B$;
18:      **end if**
19:    **end for**
20:    Update networks:
21:    Sample $N_b$ transitions from $A$, update $\theta$ using loss from (7);
22:    Sample $N_b$ transitions from $B$, update $\theta_p$ using loss from (8);
23: **end while**

---

# D   DETAILED RESULTS

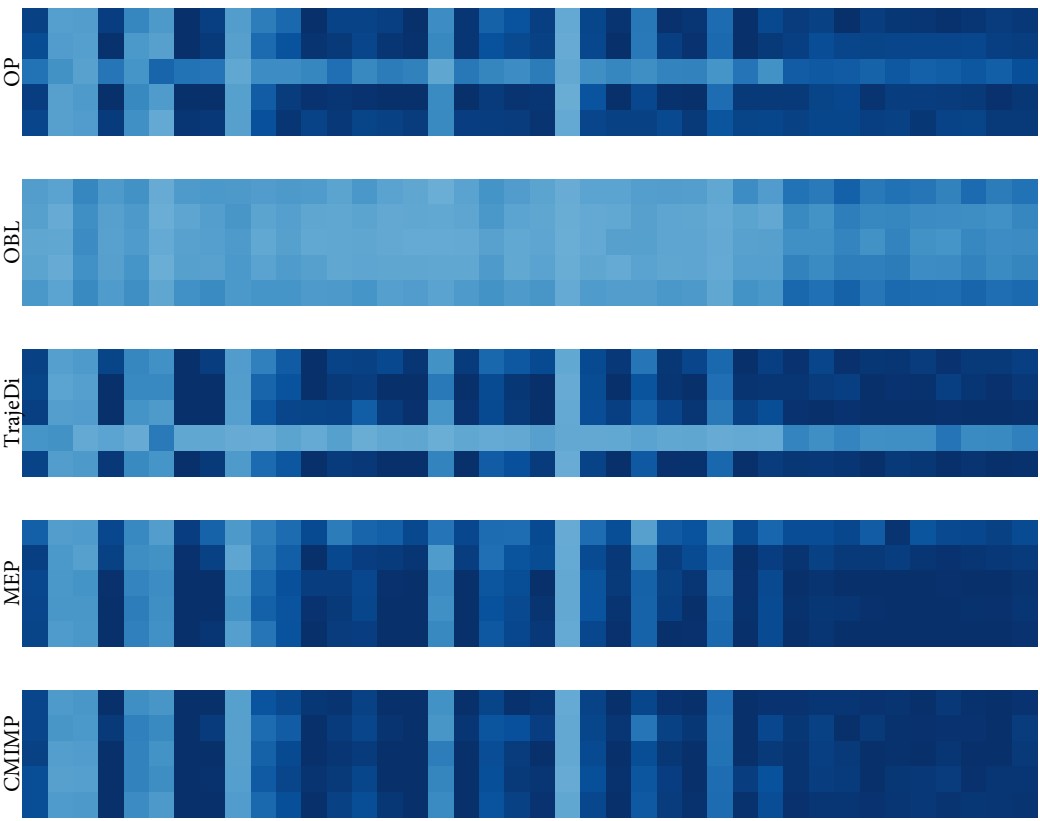

Figure 6: Detailed pair-wise 1ZSC-XP scores of all the testing frameworks.

