# OpenReview forum: "CMIMP: Effortlessly Achieving Diverse Population Training for Zero-Shot Coordination"
_ICLR.cc/2025/Conference — ICLR 2025 Conference Withdrawn Submission_

### Official Review · Reviewer_dsag · 2024-10-24

**Soundness:** 1
**Presentation:** 1
**Contribution:** 2
**Rating:** 3
**Confidence:** 3

**Summary:**

This work proposes a novel population-based training method that reduces the training cost by learning a shared "meta-policy" instead of separate individual policies. The meta-policy is a shared network with different decision heads corresponding to different agents in the population. Furthermore, it proposes a novel training objective for regularizing diversity between agents (heads) via a mutual-information based objective. The combination of these two proposal outperforms state-of-the-art methods in 2-player Hanabi.

**Strengths:**

- The architecture of the proposed "meta-policy" architecture is well motivated
- The proposed architecture allows compute-efficient training, while achieving state-of-the-art performance in 2-player Hanabi ZSC

**Weaknesses:**

- I find the paper to be generally hard to follow especially the math equations. Many variables are not defined, e.g., $N$ at line 172, $v$ in Theorem 1. Some variables are used confusingly, e.g. $u_j$ and $u_i$ in Eq.3, $N$ is Table 2. From the text, $i$ and $j$ have totally different meaning (agent index vs number of transition). $i$ and $j$ are being compared in Eq. 5 despite having different meaning.
- Table 1 is not fact-based nor scientific but a subjective judgement from the authors. I suggest the authors to either remove or justify the information with facts and citation.
- Eq. 5 needs a lot more motivation and introduction. It is not clear what is the reason we should use this specific form to approximate the mutual information
- Regarding the approximation in Eq. 5, it is not clear to me at all why taking the mean of action probability or q-values gives the approximation of the mutual information.
- The instantiation in Eq.8 gives me some understanding of the method, instead of initial the derivation, which should not be the case. Still, I find Eq. 8 to be cryptic. Partly because $i$ and $j$ are being compared despite their difference. But my rough understanding is that the objective drives different agent to have different q-values given the same observation. Being different in this case comes from the fact that the objective tries to reduce the q-values of all other agents, which is also my guess for the notation $i \neq j$, which apparently seems to well in Hanabi.
- I do not find the proposed architecture related to Meta-RL.
- The proposed training objective is supposed to "guarantee" diversity, but Eq. 8 optimizes for different Q-values which does not guarantee different behaviors.
- The paper does not ablate the inclusion of the proposed training objective. With the given architecture, I imagine that using TrajeDiv or MEP diversity constraints should also work well but is not shown in the paper.
- The paper will benefit from also running the same experiment as in Table 5 but with $\alpha=0$ to show effectiveness of the proposed method across training setups.

**Questions:**

- Can the authors elaborate why Eq. 5 is used to approximate the mutual information? How does taking the average lead to the approximation of the mutual information?
- How did the authors come up with Table 1?
- Can this method be scaled to >2 players? Would that require any changes in the architecture?

---

### Official Review · Reviewer_bWd9 · 2024-10-31

**Soundness:** 2
**Presentation:** 2
**Contribution:** 2
**Rating:** 3
**Confidence:** 3

**Summary:**

This paper studies efficient training paradigms for zero-shot-coordination (ZSC), a framework to improve the generalization performance of cooperative agents by requiring agents to coordinate with unknown agents that were not seen during training. It focuses on the use of population-based training to improve ZSC by coordinating with a population filled with diverse agents with different behaviors. Specifically, it explores the use of meta-learning for population frameworks that reduce computational costs while maintaining population diversity. It proposes a population-based zero-shot coordination framework called CMIMP. Use meta-learning with hierarchical architecture to realize a diverse population for ZSC training by maximizing conditional mutual information between actions and sub-decision modules conditioned on observation. It empirically validates the performance of CMIMP in the Hanabi environment, and conducts ablation studies on how different training modes affect the performance of population training.

**Strengths:**

- Prior work achieves population diversity by requiring agents to output differentiable action distributions, this work uses meta-learning that maximizes conditional mutual information between actions and sub-decision modules conditioned on observation. This explores the connection between meta-learning with the objective of maximizing mutual information and population training.
- Through constructing a hierarchical structure, the meta-agent trains more efficiently by reducing parameters that need to be optimized while maintaining population diversity (as evaluated by high mutual information): intuitively, task-related parameters (e.g. modules that process observation and keep history memory) can be shared and behavior-related parameters (e.g. decision-making modules) can be individually optimized.

**Weaknesses:**

- The paper lacks theoretical justification for the update method used for optimizing mutual information. In the main theoretical result, Theorem 1, it shows that one specific update rule specified in equation (6) will increase $I_j$ for some component $j$, but it does not show that the method will converge to the optimal mutual information when the update stops.

**Questions:**

- It is not clear to me why maximizing mutual information necessarily corresponds to a diverse training population. The paper states directly that a diverse population corresponds to "making the meta-agent output distinct actions with different sub-decision modules given an observation and history trajectory" which is equivalent to maximizing mutual information $I(A;U|H)$ with $A$ represents the output action, $U$ represents the index of sub-decision module, $H$ represents the observation. However, I don't see exactly how this objective necessarily maps to diverse population. Are there any connections of this objective of high mutual information to e.g. more differentiable action distributions?
- Can you given some explanation to the empirical results on page 9 which concludes that adding $\overline{I}(A;U|H)$ to the training objective at a moderate level improves ZSC, but assigning it an overly large weight have negative effect? Given it seems that the goal is to maximize diversity of training population to improve ZSC. Or is this observation due to the specific agents used during testing?

---

### Official Review · Reviewer_oYSK · 2024-11-02

**Soundness:** 2
**Presentation:** 2
**Contribution:** 2
**Rating:** 3
**Confidence:** 4

**Summary:**

The authors propose Conditional Mutual Information Maximized Population (CMIMP), a method that uses a meta-learned agent (meta-agent) to improve the efficiency of population-based training in zero-shot coordination (ZSC) tasks. Furthermore, CMIMP uses a mutual information objective to encourage diverse populations of agents to be learned by the meta-agent. The authors compare CMIMP to ZSC baselines and other population-based training methods in Hanabi.

**Strengths:**

- This paper proposes an architecture that can improve the efficiency of population-based training in ZSC tasks. Furthermore, the authors propose using a mutual information objective to encourage diverse populations of agents to be learned by their method. The combination of these two components in ZSC appears to be novel.
- Improving the efficiency of population-based training in ZSC is an important problem in MARL.

**Weaknesses:**

- The authors suggest similarities between population-based training and meta-RL, yet the connection is vague. While meta-RL focuses on fast adaptation to new tasks, CMIMP uses a multi-headed architecture that is more aligned with shared parameter frameworks than true meta-RL. A more explicit clarification on how CMIMP draws on meta-RL principles, or how it might benefit from these principles, would strengthen the narrative.
- The experimental section lacks crucial details, including the evaluation protocol, the number of random seeds, and the number of players in Hanabi (especially in Table 3). Additionally, it is unclear which algorithms were used for the 1ZSC-XP score. These omissions make it difficult to assess the results.
- The experiments are only conducted on a single environment. The authors should consider evaluating CMIMP on a broader set of environments, such as Overcooked.
- The paper does not discuss CMIMP’s sample efficiency. Evaluating how CMIMP compares to other methods in terms of sample efficiency (not just final convergence performance) would be valuable for understanding its practical viability.
- Although, E3T [1], a modern method in ZSC is mentioned in the related work, it is not included in the experiments. A comparison with E3T might be relevant to show the effectiveness of CMIMP.
- The speed and resource consumption results in Figure 3 are surprising. CMIMP is expected to be slower and require more memory with an increase in population size, because of the additional mutual information maximization objective, which requires all observations to be passed through all agent heads.
- While using mutual information as a diversity metric appears novel in ZSC, it is commonly applied in diversity maximization in MARL (e.g., [2]). Citing these related works would better position CMIMP within the existing literature.

1] Yan, X., Guo, J., Lou, X., Wang, J., Zhang, H. and Du, Y., 2024. An efficient end-to-end training approach for zero-shot human-AI coordination. Advances in Neural Information Processing Systems, 36.

2] Li, C., Wang, T., Wu, C., Zhao, Q., Yang, J. and Zhang, C., 2021. Celebrating diversity in shared multi-agent reinforcement learning. Advances in Neural Information Processing Systems, 34, pp.3991-4002.

**Questions:**

1. Why was mutual information chosen as the diversity metric? How does it compare to other diversity metrics or entropy-based measures in the context of population-based training?
2. Does CMIMP’s architecture differ from a multi-headed architecture with a mutual information objective? If so, what are the key differences?
3. Why does training time and memory consumption not significantly increase with population size in Figure 3? If the increases are minor, providing actual numbers in the appendix would add clarity.
4. What specific set of algorithms or agents was used in calculating the 1ZSC-XP score?

---

### Official Review · Reviewer_hXeu · 2024-11-11

**Soundness:** 2
**Presentation:** 2
**Contribution:** 3
**Rating:** 5
**Confidence:** 4

**Summary:**

The paper studies the problem of zero-shot coordination in RL and mentions population-based methods as a common approach to this problem. The authors address the gap that these methods suffer from inefficiency as the population size scales. The main contribution of the paper is a framework called CMIMP that consists of two main components: 1) meta-agent, and 2) mutual information regularizer.
The meta-agent has a hierarchical architecture that enables sharing parameters across the agents. The conditional mutual information is optimized as a regularizer to encourage diversity in the population. The paper proposes an alternative objective to the common unbiased estimator of the conditional mutual information and provides practical and theoretical properties of the alternative objective.
The framework is evaluated on the Hanabi environment and is compared to other population-based methods based on two different metrics for zero-shot coordination.

**Strengths:**

1) The paper is addressing an important problem in multi-agent RL.
2) The two core ideas behind the framework are interesting and valuable to study: sharing parameters across the agents, and regularization based on mutual information.
3) The authors provide an ablation study to understand the effect of parameter $\alpha$ on training. Additionally, the paper provides a comparison of different training modes. This highlights both the gains and limitations of the method.
4) The authors use two different metrics for evaluating the zero-shot coordination of CMIMP and other benchmarks. They mention the weaknesses and strengths of each score and don't solely rely on one evaluation criterion. Overall, the experiment setup for the Hanabi environment is comprehensive.

**Weaknesses:**

Methodology (Section 3)

The presentation of the problem setup in Section 3.1 requires significant revision for clarity and rigor. The meta-agent can be formally presented as a tuple containing all mentioned modules, with definitions and ranges for functions and variables. Variables $h$ and $c$ in equation 1 are used without prior definition.
Some of the preliminary terms like the $Q$-function are well-known concepts but it's better to provide a definition using your own notation as they are frequently used (in Sections 3.2 and 3.3) in presenting the core methodology of the framework.

The paper requires a more thorough explanation of how the meta-agent's hierarchical architecture is reducing the number of trainable parameters. There is only a high-level statement (lines 171-173) in this regard which is not presented well to justify the central claims of the paper.

------------------------

Experiments (Section 4)

All the results presented in the paper are only based on the Hanabi environment. The current experiment setup is good, but the gain of performance in a single environment can always be misleading.

The authors mention that the CMIMP framework is compatible with both value-based methods and policy gradient-based methods and mention two possible forms for the function $F$ in Section 3.2. The choice of value-based methods for the Hanabi environment is backed by prior work, but there are no experiments in the main section or appendix for policy-gradient-based methods.  The authors should either add experiments for policy-gradient methods such as PPO on a different environment or narrow the framework's scope to focus exclusively on value-based methods.

**Questions:**

1) Is the alternative objective $\bar{I}$ also an unbiased estimator of $I$ for both possible forms of $F$? If not, what is the bias and how does it affect the optimization?
2) Is Theorem 1 presented to prove the second property of $\bar{I}$ (line 216)? If so, please make this connection more explicit in the revision.
3) Is the source code for the experiments available?
4) Is the framework robust to various choices of hyperparameters? The paper only reports the final hyperparameters in Appendix B, with no further details about the hyperparameter tuning phase. Which hyperparameter tuning method was used for the current choices in the implementation details? RL algorithms can be sensitive to the choice of hyperparameters, and the CMIMP framework consists of multiple modules such as LSTM, thus, it's important to discuss this in more detail.

---

### Note · Authors · 2024-11-25

I have read and agree with the venue's withdrawal policy on behalf of myself and my co-authors.